# Effect of a social media-based health education program on postnatal care (PNC) knowledge among pregnant women using smartphones in Dhulikhel hospital: A randomized controlled trial

**Kalpana Chaudhary**[1]*, **Jyoti Nepal**[2], **Kusum Shrestha**[2], **Manita Karmacharya**[2], **Dipesh Khadka**[2], **Abha Shrestha**[3], **Prabin Raj Shakya**[4], **Shristi Rawal**[5], **Archana Shrestha**[1,6,7]

1 Department of Public Health, Kathmandu University School of Medical Sciences, Panauti, Nepal, 2 Department of Community Program, Dhulikhel Hospital, Kathmandu University Hospital, Dhulikhel, Nepal, 3 Department of Obstetrics and Gynecology, Dhulikhel Hospital, Kathmandu University Hospital, Dhulikhel, Nepal, 4 Biomedical Knowledge Engineering Lab, Seoul National University, Seoul, Korea, 5 Department of Clinical and Preventive Nutrition Sciences, School of Health Professions, Rutgers University, New Brunswick, NJ, United States of America, 6 Institute for Implementation Science and Health, Kathmandu, Nepal, 7 Department of Chronic Disease Epidemiology, Yale School of Public Health, New Haven, CT, United States of America

* kalpana298@gmail.com

## Abstract

### Introduction

Postnatal care services helps in detecting and subsequently managing life threatening complications. With the ubiquitous use of the mobile phone in Nepal, social media based postpartum education has the potential to increase PNC knowledge among pregnant women. This study aimed to assess the effect of social media-based health education program on PNC knowledge among pregnant women attending Dhulikhel hospital, Nepal.

### Materials and methods

We conducted a two-arm open-label randomized controlled trial among literate pregnant women visiting Dhulikhel hospital for ANC check-up from May to August, 2021. A computer-based program allocated 229 pregnant women owning smartphones with internet connectivity in a 1:1 ratio to either intervention (n = 109) or usual care (n = 120). We assessed PNC knowledge in the participants by interviewing in-person or via phone. The intervention group received a 16 minutes video on PNC and the participants were reminded to view the video every week via telephone for a month. Control group received usual care. The primary outcome of the study was change in PNC knowledge score. We utilized intent-to-treat analysis and measured the effect of the intervention on PNC knowledge score using simple linear regression analysis.

**Data Availability Statement:** All relevant data are within the paper and its Supporting Information files.

**Funding:** The author(s) received no specific funding for this work.

**Competing interests:** The authors have declared that no competing interests exist.

## Results and discussion

The mean PNC knowledge score increased by additional 8.07 points among pregnant women in the intervention group compared to the control group (95% CI: 2.35: 13.80; p-value = 0.006). The maternal care attribute knowledge increased by 4.31 points (95% CI: 1.51–7.10, p-value = 0.03) and newborn care attribute knowledge increased by 3.39 points (95% CI: 0.41–6.37, p-value = 0.02) among pregnant women in the intervention compared to the control group.

## Conclusion

A social media-based health education is effective in improving PNC knowledge score among pregnant women. Further research is needed to evaluate if this increased knowledge is translated into the increased utilization of PNC care.

## Trial registration

ClinicalTrials.gov ID: NCT05132608.

## Introduction

Globally 295,000 women died during pregnancy and childbirth in the year 2017. Among these deaths, 95% occurred in LMICs like Nepal [1]. Maternal mortality was 462 per 100,000 live births in low and lower-income countries compared to just 11 per 100,000 live births in high-income countries in 2017 [1]. About 239 women per 100,000 live births died in Nepal from pregnancy complications or childbirth in 2016; [2] 31% of deaths occurred during pregnancy, 36% at delivery or in the week after; and 33% happened 1 week to 1 year postpartum. About 60% of all these maternal deaths are preventable [3]. Difficult obstetric events lead to morbidities/disabilities in postnatal period including conditions, such as uterine prolapse, stress incontinence, hypertension, hemorrhoids, perineal tears, urinary tract infections, severe anemia, and depression [4]. Other health complications during this period are chronic pain, impaired mobility, damage to the reproductive system, genital prolapse and infertility [5]. Postnatal period is stressful for new mothers demanding emotional and psychosocial support to reduce the risk of depression [6].

About half of all postnatal deaths occur during the first week after childbirth, and majority of these deaths occur during the first 24 hours [6] due to eclampsia, postpartum hemorrhage, and puerperal sepsis [7]. Postnatal care is important to detect and prevent impairment and disabilities, and reduce serious and life-threatening complications [8]. Despite most maternal and neonatal death occurring during this postnatal period, the utilization of PNC services is low in Nepal [9]. The government of Nepal policy mandates four PNC visits (at 24 hours of delivery, the second on the third day following the delivery, the third on the seventh to the fourteenth day after delivery and the fourth on 42 days) but only 16% of women attended PNC visits as per the protocol in 2018 [10]. Only 57% of both mothers and newborns received a postnatal care check within 2 days of delivery [2].

Social media-based interventions are accessible, acceptable, and contextually modifiable; [11] and have shown promising results in reaching women of reproductive age [12] and delivering cost-effective health services in LMICs [11, 12]. Several social media-based program like Facebook, Twitter embraced across the globe to strengthen knowledge [13–16] have shown

optimistic results in the delivery of maternal, neonatal, and child health services [17–22]. Locally tailored social media and mobile based application during ANC have shown to significantly improve postnatal knowledge in Pakistan [23] and postnatal care service utilization in India [24] and community-based health and nutrition service utilization in Nepal [22].

About 73% of women (15–49 age groups) in Nepal have mobile phone subscriptions [2]. Wide use of a mobile phones in LMIC including Nepal can be potentially leveraged to increase maternal and child health outcomes through its inclusion in mobile health [25]. However, limited evidence exists in context of Nepal despite this ubiquitous, promising, and fast-growing technology-driven mobile usage in the country. Here, we aim to test the hypothesis that a social media-based postnatal health education program will improve postnatal care knowledge among pregnant women. Improved awareness and knowledge on postnatal care could lead to improved PNC practices at household and community level, with a potential to lower maternal deaths [26] and increase newborn survival [19, 27].

## Materials and methods

### Study design

This is an open-label two-arm randomized controlled trial. We allocated half of the participants to the social media-based health education, while the other half of the participants to usual care. The women in the intervention group received PNC education videos in their social media, whereas the control group received the usual services being offered in the ANC clinic at Dhulikhel hospital.

### Study site

We conducted the study at Dhulikhel Hospital- Kathmandu University Hospital -(DH-KUH) located in Dhulikhel Municipality, headquarter of the Kavre district. Dhulikhel hospital is an independent, not-for-profit, non-governmental, community-based tertiary-level hospital. Annually, more than 3000 babies are born in Dhulikhel Hospital. The catchment area of the hospital includes a population of approximately 1.9 million people from Kavrepalanchowk, Sindhupalchowk, Dolakha, Sindhuli, Ramechhap, Bhaktapur, and other surrounding districts [28].

### Participant recruitment

Between May to July 2021, 232 pregnant women were identified, and 229 eligible pregnant women were enrolled in the study. The phone numbers were obtained from ANC register at Gynecology Out Patient Department in Dhulikhel hospital. Taking reference of quasi experimental study conducted in Tanzania where change in knowledge on obstetric and newborn danger signs in the post test was 77.3% in the intervention group versus 48.0% in the control group [13], our sample size had more than 90% power to assess the effect of social media-based education on PNC knowledge [18]. A trained research assistant assessed their eligibility criteria in-person or via telephone. Inclusion criteria were i) literate pregnant women at any gestational age, ii) owned a smartphone, iii) used social media (either What's App, Viber, or Facebook) and iv) had internet connectivity on their phone (WIFI at home or mobile data). The women were excluded if they were identified with learning difficulties such as dementia or vision impairment. In this study, majority (99% of 232) of pregnant women who visited DH for ANC check-ups had mobile phones. All participants provided written or verbal informed consent.

The sample size calculation was made as: a total of 296 women, 148 in each arm were calculated to detect an effect size of 2(OR) [18] between the intervention and control arm with 59% of women having their PNC visit within 7 days of delivery [2]. The sample size calculation was based on the assumption of 80% power and 5% significance level with an attrition rate of 10 percent. The sample size was calculated using ClinCal Software [29, 30].

The 1:1 allocation was not equal. We could not enroll the original target sample size of 296 as enrollment had to be paused due to COVID-19 pandemic, leading to discrepancies in the participant's allocation to control(n = 120) or intervention(n = 109). However, this sample size had enough power (more than 90%) to assess the effect of social media-based education on PNC knowledge [18, 31].

## Randomization, allocation, concealment and blinding

A computer-based program (STATA 14) was used to randomly allocate women to either social media-based health education or usual care group. Randomization allocated women to intervention or usual care in a 1:1 ratio using simple randomization. One of the study investigators (AS) generated the random sequence and emailed to another investigator (KC), who in turn assigned participants to the intervention based on the serial number of recruitments in the random sequence using random sequence number.

## Intervention

The social media-based health education was designed using the Health Belief Model [32]. Health belief model(HBM) hypothesizes that people will take action to prevent illness if they consider themselves susceptible to a particular health condition. People tend to change health behaviors if they believe particular health condition has grave consequences in their health. We used HBM to develop intervention with the aim of making pregnant women aware of perceived susceptibility and perceived severity of postnatal care which in turn increases perceived self-efficacy [32]. HBM is a common and widely used health education approach to explain and predict preventive behaviors. HBM has been previously used to design educational intervention for improvement of oral health behavior in grade-schoolers [33] and maternal education [34]. According to the HBM as applied to PNC education, when mothers find themselves susceptible to PNC problems and understand the severity of such problems, they will be more likely to adopt recommended PNC related health behaviors. Other elements of the model, such as the balance between perceived threat and benefits, how people react to media-perceived action cues, and how well people believe they can avoid disease, all help people change harmful behaviors and adopt healthy ones. Here, PNC service utilization is referred to healthy behaviours. We plan to make follow up postpartum women to see the effect of PNC video on service utilization in coming days [35].

Initially, the investigator prepared comprehensive, pictorial and simple PowerPoint presentation on PNC guided by Health belief model with script in Nepali language. The study team including an Obstetrics-Gynecology physician reviewed the presentation on PNC. After finalizing the presentation, researcher developed the PNC video and uploaded it on the YouTube channel. The concepts used in developing the PNC educational video is summarized in Table 1.

Participants in the intervention group received a 16-minute video on general information on postnatal care including frequency, timing, places providing postnatal care, common PNC danger signs for mother and newborn, different services provided at each PNC visit, and the importance of PNC visit. The researcher sent PNC videos through instant messaging (IM) software applications such as Viber, What's App, and Facebook-messenger to the intervention

**Table 1. Concept and definition of different term of the conceptual framework included in the social media-based study.**

| Concept | Description | Action of intervention |
|---|---|---|
| **Perceived Susceptibility** | •Any woman in the postnatal period can develop postpartum complications | Pregnant women and postpartum women, their vulnerability on physical and mental health |
| **Perceived Severity** | •Problems during the postnatal period are [37]:<br>•Postpartum infections, (including uterine, bladder, or kidney functions)<br>•Excessive bleeding after delivery<br>•Pain in the perineal area (i.e., between the vagina and the rectum)<br>•Vaginal discharge<br>•Breast problems, such as swelling, infection, and clogged ducts<br>•Stretch marks<br>•Hemorrhoids and constipation<br>•Urinary or fecal incontinence<br>•Hair loss<br>•Postpartum depression<br>•Discomfort during sex<br>•Difficulty regaining your pre-pregnancy shape | Consequences in the postnatal period |
| **Perceived benefits** | •Benefits •Physical assessment including temperature, blood pressure, pulse, respiration, hemoglobin, blood group, etc.<br>•Assessment of danger Signs<br>•Any breast problems such as crack nipple, breast engorgement, breast abscess, etc.<br>•Height of fundus<br>•Cesarean section- dressing, stitching out<br>•Wound care- soakage, any discharge, hematomas, gaping, tenderness, previous Cesarean Section, •Lochia- normal/foul-smelling<br>•Assess vaginal bleeding<br>•Perineum- tear, bleeding, hematoma, swelling<br>•Taking iron/calcium<br>•Counseling for nutrition immediate family planning, immunization<br>•Anti D antibody, Rh immunoglobulin<br>•Explain Danger signs maternally/newborn: encourage for subsequent visit: advocate the importance of PNC visit.<br>•Referral [38] | Perceived benefits can be experienced by accessing PNC service as per government protocol<br>1st visit- within 24 hours<br>2nd visit within 3 days<br>3rd visit within 7 days<br>4th visit within 42 days |
| **Perceived barriers** | •Lack of knowledge of PNC, feeling of no need of PNC visit, feeling of fine, cultural barriers such as postpartum women should not go outside of the home for 7 days, geographic barriers such as geographic terrain, lack of roads; physical distance such as not within reachable distance, or with available transport;<br>•Financial barriers; quality barriers [39] | Reassurance, correction of misinformation, incentives, assistance |
| **Cues to action** | •Internal- danger signs in mother and newborn, other health problems during the postnatal period<br>•External-influence of mobile app-based education, counseling by a health worker<br>•Reminders for PNC visit | Videos on postnatal care, danger signs of mother and newborn, the importance of PNC visit, pop up PNC visit schedule |
| **Self-efficacy** | •Increase in PNC knowledge and service utilization | Visual illustration on postnatal care, danger signs of mother and newborn, the importance of PNC visit |

group. Based on the health education principle, repetitive view of the educational video would result in retention and persistence of the knowledge [36]. The pregnant women watched the PNC video for at least 4 times from enrolment to one month of intervention. The researcher assessed if the participants watched the video by asking the participants on weekly telephone calls for four weeks, relying on self-report. The detailed intervention protocol of the research is provided in S1 Data and link for video access is https://www.youtube.com/watch?v=_cJxjcJ9eIc.

## Usual care

All participants received usual care consisting of the physical assessment of maternal and fetal wellbeing, screening, treatment, and receiving preventive measures (tetanus toxoid, iron and

folate supplementation, excluding pregnancy induced hypertension and anemia, birth pre-paredness, and in-person health education, advice, and counseling (appropriate nutrition and rest, promotion of early and exclusive breastfeeding and women, smoking cessation, avoidance of alcohol and drugs) during regular ANC check-up at DH [40].

### Data collection tool and variable measurement

The list of variables, data collection techniques, tools and timeline are stated in Table 2.

### Baseline assessment

Trained research assistants interviewed the participants either in-person or via telephone (owing to the COVID-19 social distancing policy in the hospital) and assessed socio-demo-graphic and obstetric characteristics using standardized questionnaires adopted from the National Demographic Health Survey; [2] that were directly entered into an online platform developed using Kobotoolbox platform.

### PNC knowledge assessment

We adopted knowledge assessment questionnaire from a hospital-based cross-sectional study conducted to determine the knowledge of postnatal care among postpartum mothers during dis-charge in maternity hospitals in Asmara, Eritrea [31]. The Asmara study developed PNC knowl-edge questionnaire with relevant guideline prepared by WHO on post-natal care of the mother and new born [9] and former similar studies conducted in Kenya [41] and Tanzania [42]. The content validity of the questionnaire was assessed through panel of experts from Ministry of Health and Asmara College of Health Sciences and internal consistency of the tool was assessed

**Table 2. Data collection tools and schedule of different variables included in the social media-based study.**

| Description | Measurement | Tools/ Technique | Timeline |
|---|---|---|---|
| Dependent Variables | Knowledge (0–61) (20 questions, 1 score for each correct response) | Knowledge assessment questionnaire (face to face and telephone) | At baseline and end-line |
| Independent Variables | Social media (Yes, No) | Structured Questionnaire | At baseline |
| Other Variables | •Age (in years)<br>•Ethnicity (Brahmin, Chhetri, Newar, Magar, Sherpa, Kami, Others)<br>•Religion (Hindu, Buddhist, Muslim)<br>•Education (Number of years of formal education)<br>•Occupation (Service, Business, Farmer, Housewife, Student)<br>•Income (in NRs) | Structured questionnaire/ Interview (face to face and telephone) | At baseline |
| | **Obstetric variables**<br>•Gravida (one, two, others)<br>•Parity (number)<br>•Gestational week (number)<br>•ANC check-up<br>  • 4 months (Yes, No)<br>  • 6 months (Yes, No)<br>  • 8 months (Yes, No)<br>  • 9 months (Yes, No) | Structured questionnaire/ Interview (face to face and telephone) | At baseline |
| | •Planned Pregnancy (Yes, No)<br>•Mobile, Internet, and Media Use<br>•Frequency of listening radio (once a week, more than once a week, don't listen)<br>•Frequency of watching TV<br>(Once a week, more than once a week, don't watch) | Structured questionnaire/ Interview (face to face and telephone) | At baseline |

by computing Richard's Kurdson coefficient (0.75) [31]. We translated English questionnaires to Nepali language, which is the national language of Nepal and the language used by the study population. In our study context, we computed Cronbach alpha to assess the internal consistency of the tool and found to be within the acceptable range (Cronbach alpha = 0.83).

The PNC knowledge questionnaire was designed to record women's correct responses on a continuous scale. The knowledge assessment questionnaire included 20 questions (with 61 items each having one score). The questionnaire covered two components: maternal care and baby care. Maternal care component encompassed danger signs in mother (13 items), infection prevention (9 items), care of bladder (1 item), timing of sexual activity initiation (1 item), nutrition (6 items), pregnancy prevention by exclusive breastfeeding (1 item), known contraceptive methods (4 items), where to go if experience danger signs (1 item) timing of PNC visit (1 item) and necessity of PNC visit (1 item). Similarly, baby care components included how to keep baby warm (2 items), time of first newborn baby bath (1 item), umbilical care (1 item), initiation of breastfeeding (1 item), frequency of breastfeeding per day (1 item), exclusive breastfeeding (1 item), importance of vaccination (2 items), and newborn danger signs (14 items). Every correct response was given a "1" score and the incorrect response was given a "0" score. The overall response was calculated by summing up all the individual item scores. The total sum was computed twice, (1) at enrollment and (2) at the end of the study. Higher scores signified higher PNC knowledge level. An increase in the knowledge assessment scores signified an increase in the PNC knowledge.

## Follow-up assessment

We conducted follow-up interviews after 4 weeks of intervention. Data from women in both the intervention group and the control group were collected on PNC knowledge.

## Statistical analysis

**Descriptive analysis.** The descriptive statistics for baseline characteristics comparing intervention and control groups were presented as frequencies (percent) for categorical variables and means (standard deviations) for continuous variables.

**Comparison of characteristics of intervention and control group.** We conducted an independent sample t-test to compare the continuous demographic and obstetric data, including age, income, family size, and first age at menarche, a gestational week at enrollment, and number of ANC visits between the intervention and comparison groups. A chi-square test was used to compare group differences in categorical variables, including level of education, ethnicity, religion, occupation, frequency of TV and radio views, pregnancy intention, menstrual cycle, gravida, and parity.

**Primary analysis.** Primary analysis was conducted according to the intent-to-treat principle. We utilized univariate linear regression models with intervention status (yes/no) as independent variable and PNC knowledge score as the outcome.

**Testing assumptions of linear regression.** The four assumptions of linear regression: linearity, independence, normality and equal variance [43] were tested. The linearity of the relationship between social media intervention and knowledge score was determined by scatter plot of the residuals vs predicted score. The normality assumption was assessed through histograms and normal P-P plots. We looked for the residuals vs fitted line to determine equal variance. The visual inspection of these graphs showed the data were linear, normally distributed and had equal variance. By design the outcome were independent of each other as there was no matching, clustering and paired data. Therefore, all the assumptions of the independence

were fulfilled. Though all the assumptions of the linearity were met, we conducted robust measures [43] for more precision of the result estimates.

### Ethical consideration

The research was approved by the Ethical Review Board (ERB) of Kathmandu University School of Medical Sciences (IRC reference number: 106/ 20) and registered in clinicaltrials.gov (Identifier: NCT05132608). We obtained informed consent from the participant. We interviewed participant in a separate room to maintain privacy. The nature of the participation was voluntary. Any form of coercion to answer the question was forbidden.

## Results

### Sample characteristics

Fig 1 is a CONSORT diagram representing the flow of subjects through the study. We screened 232 pregnant women. A total of 229 eligible pregnant women were enrolled in the sample and randomly assigned to either intervention (social media-based education) and control group (usual care) in the study. The women not meeting the eligibility criteria (n = 9; 3.87%) or refused to give consent (n = 1; 0.43%) were excluded from the study.

All the eligible pregnant women completed the baseline interview. Of these, we were able to follow up 169 women in three months (May-August 2021). Hence, the follow-up rate of the participants was 74%. The follow up rate of participants in the control group was 83%(n = 120) and the follow up rate in the intervention was 64% (n = 109). The detailed reasons of loss to follow up of participants with respective number is mentioned in Fig 1.

### Socio-demographic characteristics

Table 3 shows the socio-demographic characteristics of the participating women. The mean age of the participants was 26 (SD = 4) years. The majority of the participating pregnant women were Hindus. The mean education years of the participants were 12 (SD = 4) years. About 61% of the participating pregnant women did not listen to the radio. In contrast, about 70% of the participating women watched TV at least once a week. The mean family size of the participants was 4.87(SD = 2.16) There were no differences in socio-demographic characteristics between the intervention and control groups at baseline.

### Obstetric characteristics

There were no differences between the groups for obstetric characteristics except for a higher gestational week (21.4 weeks± 9.30) in the intervention group versus the control group(18.3 weeks ± 9.08) (p-value = 0.04) (Table 4).

### Knowledge of danger signs of women during the postnatal period

More than two-thirds of the participants reported severe lower abdominal pain, smelly discharge from the vagina, severe headache, and excessive vaginal bleeding as the danger signs of women during the postnatal period. There was no difference in knowledge of danger signs of women during the postnatal period between the intervention and control group. (Table 5).

### Maternal care attributes knowledge

Almost all (99%) of the participants responded correctly on where to seek care if they experience any danger signs. Only about 6% of the respondents mentioned emptying the bladder

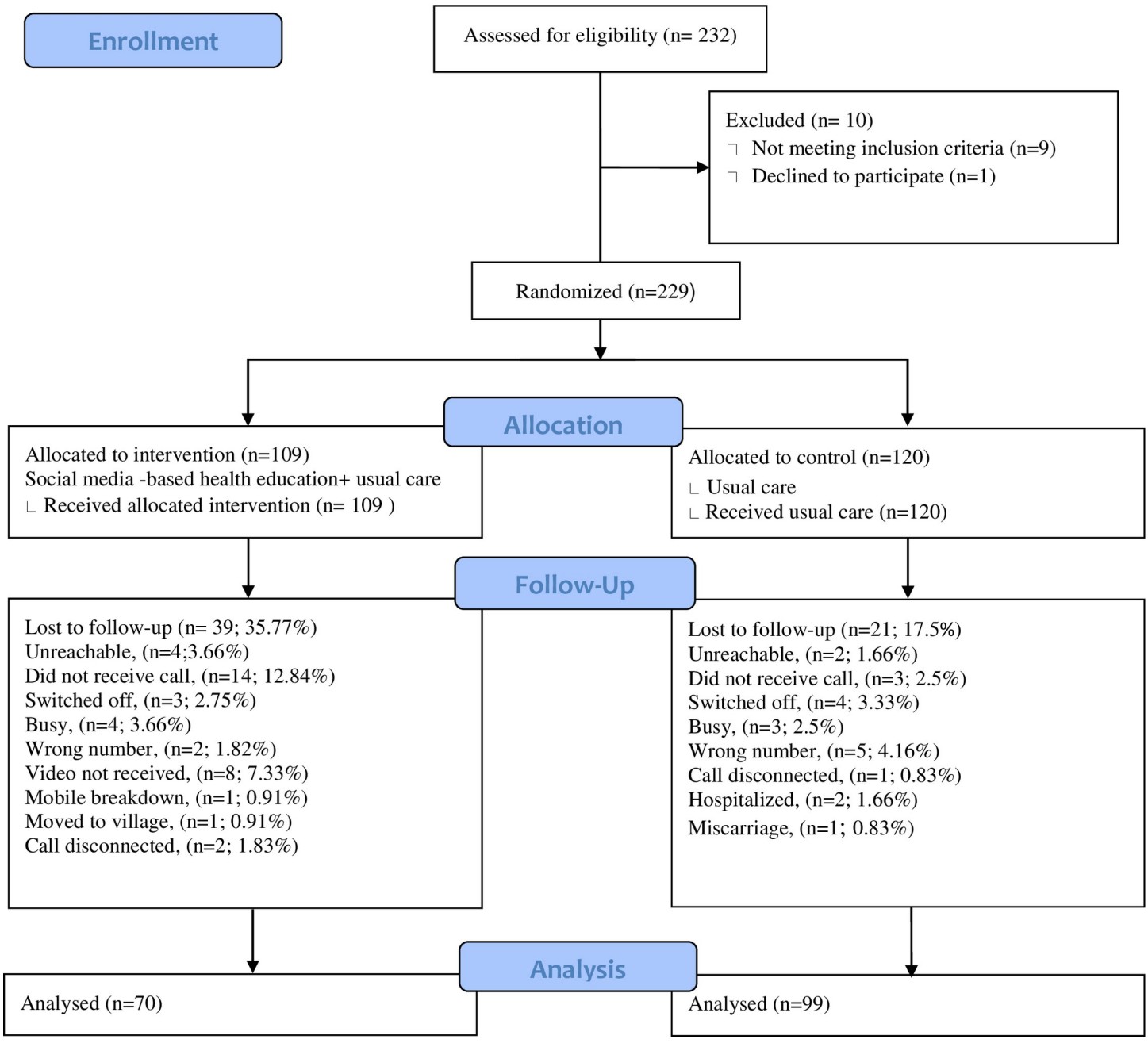

**Fig 1. CONSORT flow diagram.**

every 2 hours was the correct response for frequency of urination during postnatal period. About 9% of the respondent gave correct response for the minimum time for starting sexual intercourse was after 42 days. More than three-fourth of the participants were aware of protein, carbohydrate, vitamins and high fluid intake as necessary diet for postpartum mothers. Almost all (>90%) participants knew minilap, pills, depo and condom as the measures for family planning. There was no difference in the maternal care attributes knowledge between the intervention and control group (Table 6).

**Table 3. Socio-demographic characteristics of the study participants in intervention and control group at baseline, n = 229.**

| Characteristics | Total N = 229 n (%) | Control n = 120 n (%) | Intervention n = 109 n (%) | p-values |
|---|---|---|---|---|
| **Age in years (Mean±SD)** | 26.2± 3.93 | 26.19± 3.79 | 26.25± 4.10 | 0.90†† |
| **Ethnicity** | | | | 0.68† |
| Brahmin | 57(24.89) | 26(21.67) | 31(28.44) | |
| Chhetri/Thakuri/Sanyasi | 37(16.16) | 21(17.50) | 16(14.68) | |
| Kaami/Damai/Sarki/Gaaine/Badi | 8(3.49) | 6(5.00) | 2(1.83) | |
| Magar/Tamang/Rai/Limbu | 43(18.78) | 22(18.33) | 21(19.27) | |
| Newar | 79(34.50) | 42(35.00) | 37(33.94) | |
| Sherpa /Bhote | 1(0.83) | 1(0.83) | 0(0.00) | |
| Others | 4(1.75) | 2(1.67) | 2(1.83) | |
| **Religion** | | | | 0.43† |
| Buddhist | 24(10.48) | 15(12.50) | 9(8.26) | |
| Christian | 6(2.62) | 4(3.33) | 2(1.83) | |
| Hindu | 199(86.90) | 101(84.17) | 98(89.91) | |
| **Education years (Mean±SD)** | 12.24± 4.0) | 12.10± 3.68 | 12.39± 4.36 | 0.59†† |
| **Education Category** | | | | 0.56† |
| Primary | 17(7.42) | 8(6.67) | 9(8.26) | |
| Secondary | 54(23.58) | 31(25.83) | 23(21.10) | |
| Higher secondary | 81(35.37) | 45(37.50) | 36(33.03) | |
| Tertiary | 77(33.62) | 36(30.00) | 41(37.61) | |
| **Occupation** | | | | 0.24† |
| Business | 36(15.72) | 24(20.00) | 12(11.09) | |
| Farmer | 20 (8.73) | 9(7.50) | 11(10.09) | |
| Service | 50(21.83) | 29(24.17) | 21(19.27) | |
| Homemaker | 116(50.66) | 55(45.83) | 61(55.96) | |
| Student | 7(3.06) | 3(2.50) | 4(3.67) | |
| **Personal income (Mean±SD)** | 24001.68 ± 61571.96 | 24857.34 ± 62537.73 | 23059.66 ± 60765.37 | 0.82†† |
| **Family income (Mean±SD)** | 239429.3 ± 340483.1 | 234991.3 ± 327371.3 | 244315.2 ± 355815.6 | 0.97†† |
| **Family size (Mean±SD)** | 4.89± 2.17 | 4.78± 2.14 | 5.01± 2.20 | 0.34†† |
| **Radio** | | | | 0.79† |
| At least once a week | 68(29.69) | 36(30.00) | 32(29.36) | |
| Less than once a week | 22(9.61) | 10(8.33) | 12(11.01) | |
| Don't listen | 139(60.70) | 74(61.67) | 65(59.63) | |
| **TV** | | | | 0.21† |
| At least once a week | 161(70.31) | 80(66.67) | 81(74.31) | |
| Less than once a week | 28(12.23) | 14(11.67) | 14(12.84) | |
| Don't watch | 40(17.47) | 26(21.67) | 14(12.84) | |

†Chi-squared test

†† Students Independent t-test

## Knowledge of danger signs of the newborn

More than three-fourth of the respondent identified yellow eyes, yellow sole, unable to suckle, and fever as the danger signs of newborn. There was no difference in knowledge of danger signs of newborns between the intervention and the control groups (Table 7).

**Table 4. Obstetric characteristics of the study participants in intervention and control group at baseline, n = 229.**

| Characteristics | Total | Control | Intervention | p-value |
|---|---|---|---|---|
| | n = 229 | n = 120 | n = 109 | |
| | n (%) | n (%) | n (%) | |
| **First Age at menarche in years (Mean±SD)** | 15.18 ± 11.34 | 15.09 ± 11.09 | 15.28± 11.65 | 0.89†† |
| **Menstrual cycle** | | | | 0.07† |
| Regular cycle | 206(89.96) | 112(93.33) | 94(86.20) | |
| Irregular cycle | 23(10.04) | 8(6.67) | 15(13.76) | |
| **Gestational week (Mean±SD)** | 19.74± 9.29 | 18.27 ± 9.08 | 21.42 ± 9.30 | 0.04†† |
| **Gravida** | | | | 0.08† |
| One | 104(45.41) | 61(50.83) | 43(39.45) | |
| Two or more | 125(54.59) | 59(49.17) | 66(60.55) | |
| **Parity** | | | | 0.06† |
| Primi | 130(56.77) | 75(62.50) | 55(50.46) | |
| Multipara | 99(43.23) | 45(37.50) | 54(49.54) | |
| **Number of self-reported ANC visits** (Mean±SD) | 4.47 (1.82) | 4.43(1.76) | 4.51(1.90) | 0.74†† |

SD-Standard deviation

†- Chi-squared test

††-Student Independent t-test

## Knowledge of newborn care attributes

There was no difference in the newborn care attributes knowledge between the intervention and the control group at baseline (Table 8).

## Knowledge score of pregnant women at baseline

The overall PNC knowledge score in the intervention group was 38.02(±12.75) and, in the control, group was 37.31(±13.53) at baseline. The difference in baseline knowledge score of

**Table 5. Knowledge of danger signs in women during the postnatal period between the intervention and control group at baseline, n = 229.**

| Characteristics | Total | Control | Intervention | P-value |
|---|---|---|---|---|
| | n = 222 | n = 120 | n = 109 | |
| | n (%) | n (%) | n (%) | |
| Fever | 164(71.62) | 87(72.50) | 77(70.64) | 0.75† |
| Severe Lower abdominal pain | 161 (70.31) | 86(71.67) | 75(68.81) | 0.63† |
| Smelly discharge from the vagina | 157(68.56) | 84(70.00) | 73 (66.97) | 0.62† |
| Excessive vaginal bleeding | 179(78.17) | 89(74.17) | 90(82.57) | 0.12† |
| Urinary incontinence | 115(50.22) | 63(52.50) | 52(47.71) | 0.46† |
| Unable to control stool | 105(45.85) | 55(45.83) | 50(45.87) | 0.99† |
| Depressed | 85(37.12) | 46(38.33) | 39(35.78) | 0.69† |
| Unable to take care of self and newborn | 83(36.24) | 46(38.33) | 37(33.94) | 0.49† |
| Excessive headache | 138(60.26) | 76(63.33) | 62(56.88) | 0.31† |
| Blurred vision | 122(53.28) | 68(56.67) | 54(49.54) | 0.28† |
| High blood pressure | 134(58.52) | 73(60.83) | 61(55.96) | 0.45† |
| Fits seizures | 102(44.54) | 57(47.50) | 45(41.28) | 0.34† |
| Others | 10(4.37) | 6(5.00) | 4(3.67) | 0.62† |

†Chi-squared test

**Table 6. Maternal care attributes knowledge of intervention and control group at baseline, n = 229.**

| Characteristics | Total | Control | Intervention | p-value |
|---|---|---|---|---|
|  | n = 229 | n = 120 | n = 109 |  |
|  | n (%) | n (%) | n (%) |  |
| **Where to seek care in danger signs** |  |  |  | 0.94† |
| Health facility | 227(99.13) | 119(99.17) | 108(99.08) |  |
| Other | 2(0.87) | 1(0.83) | 1(0.92) |  |
| **Frequency of urination** |  |  |  | 0.13† |
| Every 2 hours | 13(5.68) | 4(3.33) | 9(8.26) |  |
| When felt to urinate | 71(31.00) | 34(28.33) | 37(33.94) |  |
| Don't Know | 145(63.32) | 82(68.33) | 63(57.80) |  |
| **Resume sex after delivery** |  |  |  | 0.09† |
| After 42 days | 20(8.73) | 5(4.17) | 15(13.76) |  |
| After 1 year | 10(4.37) | 4(3.33) | 6(5.50) |  |
| After 6 months | 19(8.30) | 10(8.33) | 9(8.26) |  |
| After 3 months | 44(19.21) | 23(19.17) | 21(19.27) |  |
| Other | 136(59.39) | 78(65.00) | 58(53.21) |  |
| **PNC diet** |  |  |  |  |
| Protein | 223(97.38) | 118(98.33) | 105(96.33) | 0.34† |
| Carbohydrate | 175(76.42) | 93(77.50) | 82(75.23) | 0.68† |
| Fat | 105(45.85) | 53(44.17) | 52(47.71) | 0.59† |
| Minerals | 130(56.77) | 68(56.67) | 62(56.88) | 0.97† |
| Enough fluids | 200(87.34) | 105(87.50) | 95(87.16) | 0.93† |
| Vitamins | 191(83.34) | 100(83.33) | 91(83.49) | 0.97† |
| **Menstruation delay by exclusive** |  |  |  |  |
| **breastfeeding** |  |  |  | 0.71† |
| For 3 months | 4(1.75) | 1(0.83) | 3(2.75) |  |
| For 6 months | 54(23.58) | 30(25.00) | 24(22.02) |  |
| For 1 year | 3(1.31) | 2(1.67) | 1(0.92) |  |
| For 2 years | 3(1.31) | 1(0.83) | 2(1.83) |  |
| Don't know | 165(72.05) | 86(71.67) | 79(72.48) |  |
| **Family Planning methods** |  |  |  |  |
| Minilap | 91(39.74) | 48(40.00) | 43(39.45) | 0.93† |
| Pills | 209(91.27) | 110(91.67) | 99(90.83) | 0.82† |
| Depo-Provera | 214(93.45) | 111(92.50) | 103(94.50) | 0.54† |
| Intra-uterine contraceptive device | 176(76.86) | 87(72.50) | 89(81.65) | 0.10† |
| Implant | 202(88.21) | 103(85.83) | 99(90.83) | 0.24† |
| Condom | 207(90.39) | 108(90.00) | 99(90.83) | 0.83† |
| Vasectomy | 99(43.23) | 55(45.83) | 44(40.37) | 0.40† |

†Chi squared test

Note: Numbers may not add to total due to missing of data

maternal care attributes, newborn care attributes, and the overall PNC is shown in Fig 2. There were no significant differences in the mean maternal care attributes knowledge(p-value = 0.58), newborn care attributes knowledge(p-value = 0.98) and the overall PNC knowledge(p-value = 0.73) between the intervention and control.

**Table 7. Newborn's danger signs knowledge between intervention and control group at baseline, n = 229.**

| Characteristics | Total | Control | Intervention | P-value |
|---|---|---|---|---|
| | n = 229 | n = 120 | n = 109 | |
| | n (%) | n (%) | n (%) | |
| Yellow eyes | 176(76.86) | 94(78.33) | 82(75.23) | 0.57† |
| Yellow hand and palm | 147(64.19) | 77(64.17) | 70(64.22) | 0.99† |
| Yellow sole | 170(74.24) | 89(74.17) | 81(74.31) | 0.98† |
| Umbilical problem | 117(51.09) | 66(55.00) | 51(46.79) | 0.21† |
| Eye problem | 111(48.47) | 63(52.50) | 48(44.04) | 0.20† |
| Unable to breastfeed/suckle | 176(76.86) | 93(77.50) | 83(76.15) | 0.80† |
| Convulsion/Seizure | 104(45.41) | 59(49.17) | 45(41.28) | 0.23† |
| Fever | 177(77.29) | 91(75.83) | 86(78.90) | 0.58† |
| Difficulty breathing | 154(67.25) | 82(68.33) | 72(66.06) | 0.71† |
| Lethargic | 111(48.47) | 60(50.00) | 51(46.79) | 0.62† |
| Fussy/Irritable | 142(62.01) | 78(65.00) | 64(58.72) | 0.32† |
| Abdominal distension | 144(62.88) | 82(68.33) | 62(56.88) | 0.07† |
| Severe vomiting | 154(67.25) | 88(73.33) | 66(60.55) | 0.04† |
| Others | 51(22.27) | 25(20.83) | 26(23.85) | 0.58† |

†Chi-squared test

## Effect of social media-based education on PNC knowledge score of pregnant women

The difference in score of maternal care attributes knowledge, newborn care attributes knowledge and the overall PNC knowledge is shown in Fig 3. The knowledge score of intervention group was higher than control group in maternal care attributes (p-value <0.001), newborn care attributes (p-value <0.001) an overall PNC knowledge (p-value <0.001).

Table 9 shows the result of linear regression model assessing effect of social media-based intervention on PNC knowledge. PNC knowledge score increased significantly among pregnant women in the intervention group compared to the control group. The maternal care attribute knowledge increased by additional 4.31 scores (95% CI: 1.51–7.10,), newborn care attribute knowledge increased by additional 3.39 scores (95% CI: .41–6.37) and overall PNC knowledge score increased by additional 8.07 scores (95% CI: 2.35–13.80) among pregnant women in the intervention compared to the control group.

## Discussion

This study determines that tailored social media-based health education guided by health belief model are effective in increasing PNC knowledge score among pregnant women attending DH for antenatal check-up. The mean maternal care attribute knowledge increased by 4.31 scores, newborn care attribute knowledge increased by 3.39 scores, and overall PNC knowledge score increased by 8.07 scores among pregnant women in the intervention compared to the control group. The maternal care attributes knowledge included knowledge of danger signs, place to seek care if experiencing danger signs, frequency of urination in postnatal period, initiation of sex after delivery, PNC diet, delay in menstruation by exclusive breastfeeding, family planning methods, and PNC visit schedule. The newborn care attributes knowledge included knowledge of danger signs of newborns, keeping baby warm, first newborn bathing, cord care, first breastfeeding initiation after birth, breastfeeding frequency in 24 hours, exclusive breastfeeding duration, vaccine and importance of vaccine to newborns. The increased

**Table 8. Newborn care attributes knowledge of pregnant women between the intervention and control group at baseline, n = 229.**

| Characteristics | Total | Control | Intervention | p-value |
|---|---|---|---|---|
| | n = 229 | n = 120 | n = 109 | |
| | n (%) | n (%) | n (%) | |
| **Keeping baby warm** | | | | 0.45† |
| Skin to skin contact | 2(0.87) | 2(1.67) | 0(0.00) | |
| Wrap the baby with a cloth | 132(57.64) | 65(54.17) | 67(61.47) | |
| Both of the above | 75(32.75) | 42(35.00) | 33(30.28) | |
| Others | 19(8.30) | 10(8.33) | 9(8.26) | |
| Don't know | 1(0.44) | 1(0.83) | 0 (0.00) | |
| **First newborn bathing** | | | | 0.27† |
| Immediately | 6(2.62) | 5(4.17) | 1(0.92) | |
| After 24 hours | 94(41.05) | 46(38.33) | 48(44.04) | |
| After one week | 6(2.62) | 3(2.50) | 3(2.75) | |
| Other | 29(12.66) | 12(10.00) | 17(15.60) | |
| Don't know | 94(41.05) | 54(45.00) | 40(36.70) | |
| **Cord care** | | | | 0.79† |
| Keep cord clean and dry | 106(46.29) | 53(44.17) | 53(48.62) | |
| Others | 33(14.21) | 18(15.00) | 15(13.76) | |
| Don't know | 90(39.30) | 49(40.83) | 41(37.61) | |
| **Breastfeeding initiation after birth** | | | | 0.86† |
| Within 30 minutes after birth | 30(13.10) | 15(12.50) | 15(13.76) | |
| Within 1 hour after birth | 63(27.51) | 31(25.83) | 32(29.36) | |
| Within 24 hours after birth | 4(1.75) | 3(2.50) | 1(0.92) | |
| Others | 22(9.61) | 12(10.00) | 10(9.17) | |
| Don't know | 110(48.03) | 59(49.17) | 51(46.71) | |
| **Breastfeeding frequency in 24 hours** | | | | 0.33† |
| At least 8 times a day | 30 (13.10) | 14(11.67) | 16(14.68) | |
| When the baby cries | 37(16.16) | 20(16.67) | 17(15.60) | |
| Others | 47(20.52) | 20(16.67) | 27(24.77) | |
| Don't know | 115(50.22) | 66(55.00) | 49(44.95) | |
| **Exclusive breastfeeding duration** | | | | 0.34† |
| For the first 3 months | 1(0.44) | 0(0.00) | 1(0.92) | |
| For the first 4 months | 1(0.44) | 1(0.83) | 0(0.00) | |
| For the first 6 months | 188(82.10) | 103(85.83) | 85(77.98) | |
| Other | 5(2.18) | 2(1.67) | 3(2.75) | |
| Don't know | 34(14.85) | 14(11.67) | 20(18.35) | |
| **Vaccine** | 229(100.00) | 120(100.00) | 109(100.00) | |
| **Importance of vaccine** | | | | 0.84† |
| | | | | 0.60† |
| To prevent disease | 213(93.01) | 112(93.33) | 101(92.66) | |
| Others | 5(2.18) | 2(1.67) | 3(2.75) | |
| Don't know | 11(4.80) | 6(5.00) | 5(4.59) | |
| **PNC Visit schedule** | | | | |
| Don't know | 221(98.22) | 115(97.46) | 106(99.07) | |
| Within 24 hours, 3 days, 7 days of birth | 1(0.44) | 1(0.85) | 0(0.00) | |
| Within 24 hours, 7 days, 10 days of birth | 2(0.89) | 1(0.85) | 1(0.93) | |
| Within 7 days, 10 days, 30 days of birth | 1(0.44) | 1(0.85) | 0(0.00) | |

†Chi-squared test

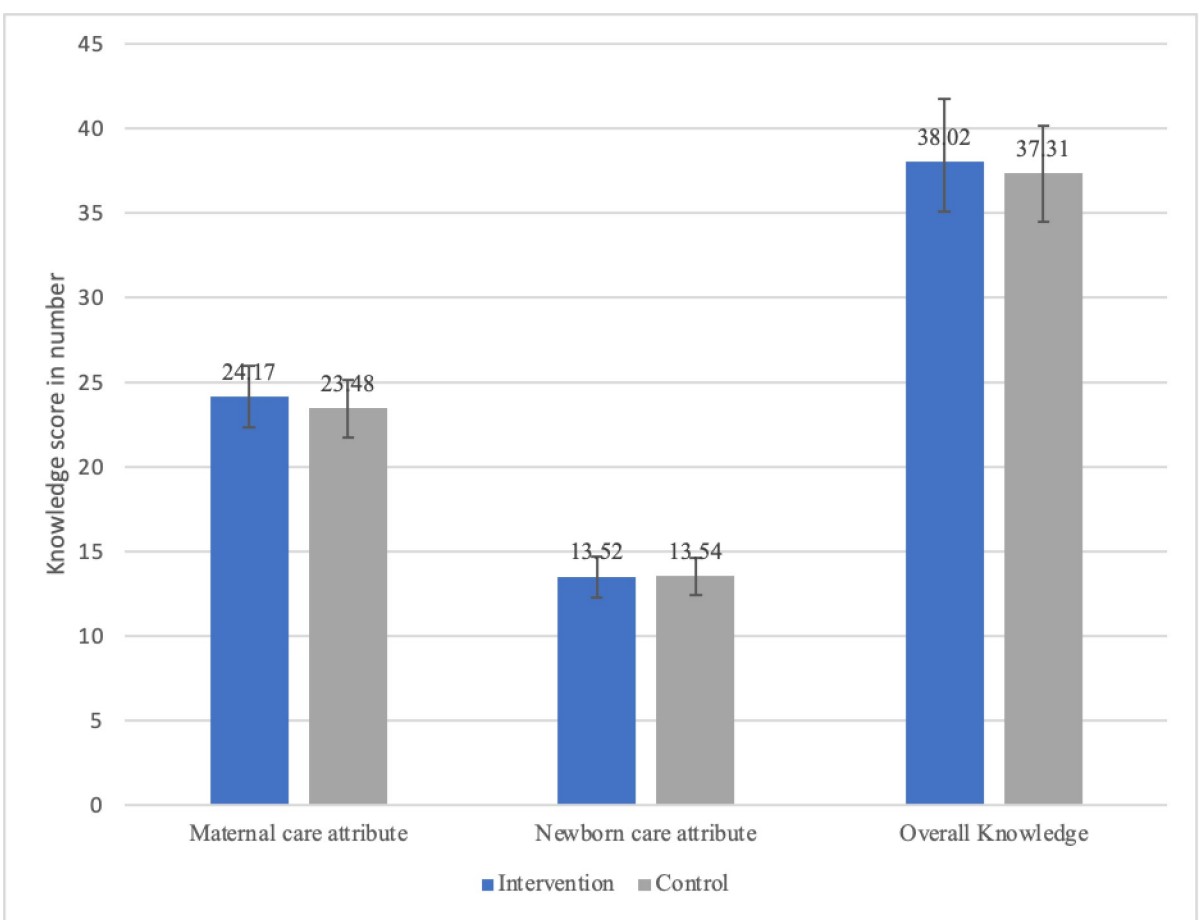

**Fig 2. Bar graph showing PNC knowledge score of intervention and control group at baseline.**

maternal and newborn care attributes knowledge develops confidence among women to take care of themselves and their babies, adapt to new conditions [44] and to go through the post-partum period successfully [45] and also increase PNC service utilization [6]. The overall PNC knowledge addresses delay in recognition of complications, delay in reaching appropriate care, and delay in receiving appropriate care [6].

The total PNC knowledge score increased by 8 points among intervention group compared to control group, which is 23% higher score among intervention group compared to control group. The maternal care attribute score was increased by 4.3 points, which is 18% higher score in intervention group compared to control group; and newborn care attribute score was increased 3.39 points, which is 32% higher in intervention group compared to control group. Women's engagement in the social media such as Facebook, Instagram, Twitter etc. enhanced the feasibility of disseminating health information to the participants at targeted locations in other settings as well [46]. The information provided through mobile phones were acceptable to the pregnant women [47]. Pregnant women are receptive and willing to make use of tech-nology based health education in order to improve their health [48].

This mobile-phone based intervention's positive effect is plausible as the unique character-istics of the mobile phone are ubiquity, mobility, constant availability, and multiple media modalities, among others [48, 49]. In addition, mobile phones reduce the feeling of being observed and the participants can learn at their own pace in their leisure time [49]. Women

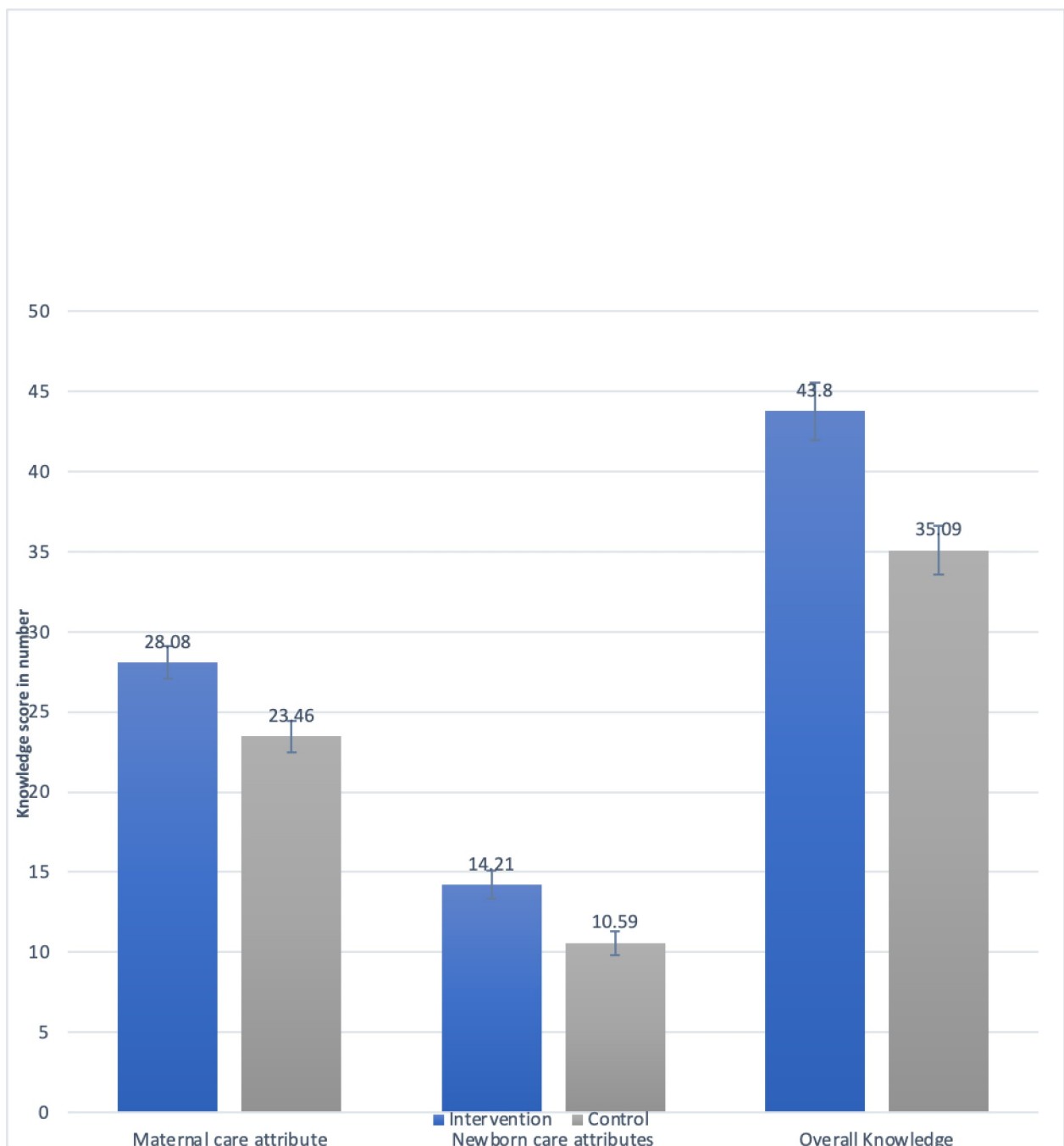

**Fig 3. Bar graph showing PNC knowledge score of intervention and control group at end-line.**

are motivated to learn when they receive educational messages that are interesting, easy to grasp, and aligned with their physiological state during pregnancy and postpartum [50]. These positive attributes of mobile phone including ease-of use and familiarity [48] could have potentially contributed to enhanced PNC knowledge among the participants assigned to the social media-based PNC education in our study.

**Table 9. Effect of social media-based education on PNC knowledge score of pregnant women attending ANC check-up at DH hospital in the intervention group, n = 67.**

| Characteristics | Intervention | | |
|---|---|---|---|
| | β Coefficient | 95% CI | p-value |
| Maternal care attributes score | 4.31 | 1.51–7.10 | 0.003 |
| Newborn care attributes score | 3.39 | .41–6.37 | 0.02 |
| Total knowledge score | 8.07 | 2.35–13.80 | 0.006 |

The positive direction of effect of intervention found in our study agrees with other experimental studies conducted in similar study settings. In a controlled quasi experimental study conducted among pregnant women attending hospital for antenatal care in Tanzania, interactive alert mobile messaging system of receiving and sending messages reported 29% (48% to 78%) increase in knowledge of danger signs of pregnancy in the intervention group compared to the control group in a sample of 450 women. Urban residence of literate pregnant women with increasing phone ownership were the predictors for increasing knowledge on danger signs of pregnancy [49]. The interactive messaging system that offered two-way communication whereby participants could send and receive health education messages led to positive effect of the intervention.

In a randomized controlled trial study conducted in Indonesia, counseling aided with mobile application named Suami Siaga plus found 20% increment in husband's score in the knowledge of danger signs in the intervention compared to the control group. The possible explanation of increase in the knowledge score suggested methods of health education such as social media had an influence on participants knowledge score. Mobile app-based health education aided with counseling yielded better effect in increasing knowledge score [15]. In a study by Parisa, the mean health literacy score after the intervention increased by 12.68 ± 6.31 (mean score = 58.03 ± 6.57) score in the experimental group and 0.21 ± 4.25 (mean score = 42.23 ± 8.47) in the control group from 45.35 ± 6.75 and 42.02 ± 7.41 score in the experimental and control group respectively at baseline. Compared to the present study, higher change in the mean knowledge score after mobile app training was attributed by the duration of the intervention (8 weeks) and the nature of the intervention where the participants viewed the educational material in the mobile app, and also had access to researcher's cellphone number to contact if she had any questions about the app. Also, the change in mean knowledge score in the intervention group was influenced by age of women and spouse, education level of women and spouse, occupation of women and spouse, monthly income, and place of residence [51].

In contrast to the findings from our study, pseudo randomized controlled trial in India with similar intervention (audio messages included messages on maternal and new born care; and messages were sent twice a week) and conducted among 2016 pregnant women found no significant changes in knowledge of pregnant women in the intervention group. The findings of this study suggested knowledge indicators measured in this study were not directly related to the specific messages or practices. The participants of the study already possessed high level of knowledge on some aspects of maternal care. About two-third of the pregnant women had already given birth to babies making them extremely familiar with the topics covered in the maternal care knowledge [52].

Even through there was about 35.77% attrition rate, the characteristics of the intervention and control participants who were loss to follow up were similar. Assessment in the characteristics of intervention and control group at end-line showed good balance eliminating the chance of potential confounding by attrition. Our study design was open masked leading to

the possibility of contamination of information between the intervention and control groups as the participants were taken from the same study site. The contamination of information might have underestimated the result estimates.

The findings of the study have important implications in terms of future research. PNC knowledge is associated with increased PNC service utilization [53]. Findings of different interventional studies reported increase in PNC service utilization with enhanced PNC knowledge [18–21]. Further research with longer follow-up duration is essential to determine the effect of increased PNC knowledge score in PNC service utilization in our setting. Once the effect of social media-based PNC education on PNC service utilization is determined, it is possible to design tailored, culturally appropriate, targeted social media-based educational intervention to increase PNC service utilizations. Also, other interventional studies are recommended to test the effect of this intervention in rural areas for scaling up the intervention to other settings as well and the region at large.

Our study has many strengths. In Nepal, this is one of the few studies with randomized controlled design conducted to assess the effect of social-media based education in increasing the PNC knowledge of pregnant women. As a randomized controlled design study, with strongest empirical evidence it was possible to establish temporal relationship and hence infer causal relationship between intervention and primary outcome, PNC knowledge. The use of standardized knowledge assessment questionnaire and robust measures in analyzing data were the strengths of this study. Use of health belief model for developing social media-based PNC education was another strength of this study. This study provided new evidences that social media-based education program is effective in creating awareness about different issues on post-natal care as compared to conventional antenatal care health education provided in ANC clinic at DH.

There are several limitations of the study. First, the short follow -up duration of the study participants which could have overestimated the effect of intervention in increasing PNC knowledge score among pregnant women in the intervention group. Second, inclusion of literate pregnant women owning smartphones with access to internet in the study limits the generalizability of the study. Also, the study was conducted in an urban setting. Therefore, its generalizability to rural settings cannot be confirmed. Third, use of social media for intervention which limited the PNC education and can limit its implications to those who could afford it. Women not having access to internet were deprived from the advantage of social media-based education and hence had to rely on the conventional health education program. Fourth, the assessment of video views by pregnant women was self-reported. However, we tracked the number of views on the YouTube—there were about 701 views on the Youtube videos for 167 intervention participants. And finally, some attrition of participants over the 1-month follow-up period were likely due to the use of telephone calls for follow- up of the participants in the study affecting the precision of the estimates.

## Conclusion

This study determines that social media-based education is an innovative and promising approach in increasing PNC knowledge in the sample of pregnant women visiting DH for ANC check-up. The PNC knowledge score showed positive significant effect with the social media-based education. These findings provide important evidence regarding the potential to include social media-based education to disseminate information on any maternal care issues like PNC in a low resource setting like Nepal with ubiquitous smartphone users, where increase in PNC knowledge may have substantial effect in enhancing maternal and neonatal health. The findings of the study suggest the need of further research with longer follow-up duration to determine the effect of increased PNC knowledge score in PNC service utilization.

Once the effect of social media-based PNC education on PNC service utilization is determined, it is possible to design tailored, culturally appropriate, targeted social media-based educational intervention to increase PNC service utilization and hopefully contribute towards achieving the targets of Sustainable Development Goals in maternal health. We expect that this study will serve as a basis for evidence-based policy and program development and implementation with regards to inclusion of social media in maternal and neonatal health.

## Supporting information

**S1 Checklist. CONSORT 2010 checklist of information to include when reporting a randomised trial**\*.
(DOC)

**S1 Appendix. Intervention Protocol.**
(DOCX)

**S1 Fig. Intervention matrix.**
(TIF)

**S1 Table. Factors associated with knowledge change of participants in the intervention and control group.**
(TIF)

**S1 Data.**
(CSV)

## Acknowledgments

We acknowledge Obstetrics and Gynecology department, Dhulikhel hospital, Department of Public Health, Kathmandu University School of Medical Sciences for their guidance and support. We are grateful to Bikram Adhikari for assisting with manuscript submission and participants for contributing their time to participate in this study.

## Author Contributions

**Conceptualization:** Kalpana Chaudhary, Archana Shrestha.

**Data curation:** Kalpana Chaudhary, Kusum Shrestha, Manita Karmacharya.

**Formal analysis:** Kalpana Chaudhary.

**Investigation:** Kalpana Chaudhary, Archana Shrestha.

**Methodology:** Kalpana Chaudhary, Shristi Rawal, Archana Shrestha.

**Project administration:** Kalpana Chaudhary, Jyoti Nepal.

**Resources:** Kalpana Chaudhary, Abha Shrestha, Shristi Rawal.

**Supervision:** Archana Shrestha.

**Validation:** Kalpana Chaudhary, Archana Shrestha.

**Visualization:** Kalpana Chaudhary.

**Writing – original draft:** Kalpana Chaudhary.

**Writing – review & editing:** Jyoti Nepal, Kusum Shrestha, Manita Karmacharya, Dipesh Khadka, Abha Shrestha, Prabin Raj Shakya, Shristi Rawal, Archana Shrestha.

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
