## [Decision Letter · Decision Letter 0]

3 Aug 2022

PONE-D-22-03664

Effect of a Social media-based Health Education Program on Postnatal care (PNC) Knowledge among Pregnant Women using Smartphones in Dhulikhel Hospital: A Randomized Controlled Trial

PLOS ONE

Dear Dr. Chaudhary,

Thank you for submitting your manuscript to PLOS ONE. After careful consideration, we feel that it has merit but does not fully meet PLOS ONE’s publication criteria as it currently stands. Therefore, we invite you to submit a revised version of the manuscript that addresses the points raised during the review process.

The manuscript has been evaluated by three reviewers, and their comments are available below.

The reviewers, in particular reviewer two, have raised a number of concerns that need attention. They have a number of queries on the intervention approach, the applicability of the Health Belief Model, and the similarity of results between control and intervention groups. They also note that a comparative analysis between participants in terms of demographics, education or other criteria could be valuable. 

Could you please revise the manuscript to carefully address the concerns raised?

We look forward to receiving your revised manuscript.

Kind regards,

Alice Coles-Aldridge

Editorial Office

PLOS ONE

Journal Requirements:

Reviewers' comments:

Reviewer's Responses to Questions

**Comments to the Author**

1. Is the manuscript technically sound, and do the data support the conclusions?

Reviewer #1: Yes

Reviewer #2: Partly

Reviewer #3: Yes

2. Has the statistical analysis been performed appropriately and rigorously? 

Reviewer #1: Yes

Reviewer #2: Yes

Reviewer #3: Yes

3. Have the authors made all data underlying the findings in their manuscript fully available?

Reviewer #1: Yes

Reviewer #2: Yes

Reviewer #3: No

4. Is the manuscript presented in an intelligible fashion and written in standard English?

Reviewer #1: Yes

Reviewer #2: Yes

Reviewer #3: Yes

5. Review Comments to the Author

Reviewer #1: The article has clearly reported the findings that are significant in addressing maternal and child health, a major global public health concern. All the section are well articulated. Therefore the paper can be accepted after the minor corrections are done.

Comments to the authors

Title: Clear and specific to the concept under study

Abstract: The introduction statement on background (Postnatal care (PNC)” can detect and subsequently manage life threatening complications” (need to be rephrased since PNC is a service and it can’t detect or manage rather it is through the service that this can be done. Methods: Looks okay, but since the ratio of allocating group was 1:1, then the numbers should not have differed much 109 vs 120. Results and conclusion are well articulated.

Introduction: The section has well described background of the area under the study. Grammatical errors such as in the year 2017 instead of in 2017. Low- and middle-income countries can be abbreviated. Delete the word resulting in this sentence “Nepal resulting from pregnancy complications or childbirth in 2016”. Line 96: The statement need revision “Several social media-based program like Facebook, Twitter embraced across the globe to strengthen knowledge and delivery of maternal, neonatal, and child health services”. Line 97: Correct the word improved to improve. Line 99: remove the word subsequently, since this is not in Pakistan which is the country referred to in the preceding statement.

Methods: The section is well elaborated but there is need to correct grammatical errors. Line 104, learning difficulties (dementia), this can be written as, such as dementia, putting it in brackets gives this a different meaning. Line 172:correct the word reminder. Line 173; remove “the” in before the making. Line 203, remove “a” in “relevant a guideline”. Line 219 add the content for (1 item). Line 222, remove 2in” from “in the”. Line 225-226, need editing for the statement “The total sum was 226 computed one during the recruitment and the other at the end of the study”. Though Health belief model is referred to in this study, it is not clear how it was used and how the its components were related or integrated into the concept of social media and PNC. Apart from ethical clearance, were there other ethical considerations. How was rigor maintained.

Results: The section is well done analysis and good presentation of the findings. In the consortium flow diagram, it is not clear why wrong number is tabulated twice in follow up.

Discussion: Line 413, I suggest you change the statement hence pass the critical postpartum successfully to go through the postpartum period successfully. Line 460 (The univariate analysis reported social media-based education significantly associated with their PNC knowledge score) is not clear.

The discussion has only compared findings with similar setting highlighting Tanzania, Indonesia and India. The study in India does not elaborate on what type of intervention, therefore not clear if they are comparable.

The references need editing since some are incomplete and wrongly done.

Reviewer #2: This manuscript presents a randomized controlled trial to measure the effectiveness of a social media-based Health Education Program on Postnatal Care (PNC) among Nepali women. Postnatal care is often considered as a neglected reproductive health service in many countries which highly influences the morbidity and mortality rate among new mothers and newborns. Therefore, this manuscript falls into an area where more research is highly needed and appreciated.

In an attempt to promote awareness and PNC education among pregnant women, a number of pregnant women have participated in the intervention that consists of watching a 16-min video elaborated by the authors on different aspects of PNC throughout a period of one month. The authors state that the participants in the intervention group received weekly reminders to watch the video via phone calls. The control group, on the other hand, was given the usual antenatal care.

Overall, the manuscript is well-written and well-structured. However, there are multiple weaknesses on this study, particularly in terms of the intervention approach.

First, I cannot discern the usefulness of having to watch the ‘same non-changing’ video on a weekly basis during that one-month of intervention. Also, how did you make sure that the video has been watched more than once? The issue in this study is that we do not get the real objective of the study; whether is it measuring the effectiveness of promoting PNC awareness through a video or measuring the efficacy of the video to disseminate information about PNC? I believe that the authors are akin to promote PNC education but I think the approach adopted is quite weak and inadequate.

Another issue worth mentioning is the use of the Health Belief Model. This model was developed to help understand and predict health behaviors. I do not see its applicability in the present study. The authors reported that the video has been designed using the HBM in a way that the video would likely highlight the risks and benefits that new mothers will get during their postnatal period. The authors are advised to review the utilization of HBM.

Plus, according to the statistical results provided, there is no substantial effectiveness perceived in raising awareness about PNC among the participants through the video’s content. The analysis shows quite similar results between the control and intervention groups.

I sincerely value the time and effort deployed in this study and I do recommend the authors to exploit the results they have obtained differently to give more strength to their study. A comparative analysis between participants in terms of demographics or education or other criteria, could be valuable.

Reviewer #3: A two-arm randomized controlled clinical trial was conducted which aimed to assess the effect of an intervention of a social media-based health education program on postnatal care knowledge among pregnant women at a single institution in Nepal. Postnatal care knowledge score, maternal care attribute knowledge, and newborn care significantly increased in the intervention arm compared to the control arm.

Minor revisions:

1- Carefully proofread the abstract, paying attention to the placement of commas.

2- The standard statistical term for average is mean.

3- State and justify the study’s target sample size with a pre-study statistical power calculation. The power calculation should include: (1) the estimated outcomes in each group; (2) the α (type I) error level; (3) the statistical power (or the β (type II) error level); (4) the target sample size and (5) for continuous outcomes, the standard deviation of the measurements.

4- Line 240: Replace “in” with “as.”

5- Indicate if the continuous variables presented in the tables were checked for normal distributions prior to applying the t-tests.

6- Lines 367-8: Provide measures of dispersion for the values 38.02 and 37.31.

7- Bar graph figures: Label the y-axes.

8- Figure S3: Overall is one word. Knowledge is misspelled.

6. PLOS authors have the option to publish the peer review history of their article (what does this mean?). If published, this will include your full peer review and any attached files.

Reviewer #1: No

Reviewer #2: No

Reviewer #3: No

---

## [Author Response · Author response to Decision Letter 0]

23 Oct 2022

Thank you for giving us the opportunity to revise and resubmit this manuscript. I appreciate the time and effort provided by the reviewer. I have incorporated the suggested changes into the manuscript to the best of my ability. The manuscript has certainly benefited from these insightful suggestions. I look forward to working with you and the reviewer to move this manuscript closer to publication in the Journal of PLOS ONE.

---

## [Decision Letter · Decision Letter 1]

5 Jan 2023

Effect of a Social media-based Health Education Program on Postnatal care (PNC) Knowledge among Pregnant Women using Smartphones in Dhulikhel Hospital: A Randomized Controlled Trial

PONE-D-22-03664R1

Dear Dr. Chaudhary,

We’re pleased to inform you that your manuscript has been judged scientifically suitable for publication and will be formally accepted for publication once it meets all outstanding technical requirements.

Kind regards,

Miquel Vall-llosera Camps

Senior Editor

PLOS ONE

Reviewers' comments:

Reviewer's Responses to Questions

**Comments to the Author**

1. If the authors have adequately addressed your comments raised in a previous round of review and you feel that this manuscript is now acceptable for publication, you may indicate that here to bypass the “Comments to the Author” section, enter your conflict of interest statement in the “Confidential to Editor” section, and submit your "Accept" recommendation.

Reviewer #1: All comments have been addressed

Reviewer #2: All comments have been addressed

Reviewer #3: (No Response)

2. Is the manuscript technically sound, and do the data support the conclusions?

Reviewer #1: Yes

Reviewer #2: Yes

Reviewer #3: Yes

3. Has the statistical analysis been performed appropriately and rigorously? 

Reviewer #1: Yes

Reviewer #2: Yes

Reviewer #3: Yes

4. Have the authors made all data underlying the findings in their manuscript fully available?

Reviewer #1: Yes

Reviewer #2: Yes

Reviewer #3: Yes

5. Is the manuscript presented in an intelligible fashion and written in standard English?

Reviewer #1: Yes

Reviewer #2: Yes

Reviewer #3: Yes

6. Review Comments to the Author

Reviewer #1: The comments raised have been adequately addressed in addition the Conclusions are presented in an appropriate fashion and are supported by the data.

Reviewer #2: (No Response)

Reviewer #3: Minor Revisions:

1- The standard statistical term for average is mean. Lines 50, 428, and the abstract still contain the term average.

2- Line 149: Refer to the software as "ClinCalc" software.

7. PLOS authors have the option to publish the peer review history of their article (what does this mean?). If published, this will include your full peer review and any attached files.

Reviewer #1: No

Reviewer #2: No

Reviewer #3: No

---

## [Editor Report · Acceptance letter]

10 Jan 2023

PONE-D-22-03664R1 

Effect of a Social media-based Health Education Program on Postnatal care (PNC) Knowledge among Pregnant Women using Smartphones in Dhulikhel Hospital: A Randomized Controlled Trial 

Dear Dr. Chaudhary:

I'm pleased to inform you that your manuscript has been deemed suitable for publication in PLOS ONE. Congratulations! Your manuscript is now with our production department. 

Kind regards, 

on behalf of

Dr. Miquel Vall-llosera Camps 

Staff Editor

PLOS ONE